# Interstitial Photodynamic Therapy for Glioblastomas: A Standardized Procedure for Clinical Use

**DOI:** 10.3390/cancers13225754

**Published:** 2021-11-17

**Authors:** Henri-Arthur Leroy, Gregory Baert, Laura Guerin, Nadira Delhem, Serge Mordon, Nicolas Reyns, Anne-Sophie Vignion-Dewalle

**Affiliations:** 1Department of Neurosurgery, CHU Lille, F-59000 Lille, France; Nicolas.reyns@chru-lille.fr; 2INSERM, CHU-Lille, U1189-ONCOTHAI-Assisted Laser Therapy and Immunotherapy for Oncology, Univ-Lille, F-59000 Lille, France; Gregory.baert@inserm.fr (G.B.); laura.guerin@inserm.fr (L.G.); nadira.delhem@inserm.fr (N.D.); serge.mordon@inserm.fr (S.M.); anne-sophie.vignion@inserm.fr (A.-S.V.-D.)

**Keywords:** interstitial, photodynamic therapy, glioblastoma, brain tumor, treatment planning system

## Abstract

**Simple Summary:**

The most frequent primary high-grade brain tumors are glioblastomas (GBMs). The current standard of care for GBM is maximal surgical resection followed by radiotherapy and chemotherapy. Despite all these treatments, the overall survival is still limited, with a median of 15 months. The challenge is to improve the local control of this infiltrative disease. Interstitial photodynamic therapy (iPDT) is a minimally invasive treatment relying on the interaction of light, a photosensitizer and oxygen. It consists of introducing optical fibers inside the tumor to illuminate the cancer cells which have been sensitized to light thanks to a natural photosensitizer agent. Herein, we propose a standardized and reproducible workflow for the clinical application of iPDT to GBM. This workflow, which involves intraoperative imaging, a dedicated treatment planning system (TPS) and robotic assistance for the implantation of stereotactic optical fibers, represents a key step in the deployment of iPDT for the treatment of GBM.

**Abstract:**

Glioblastomas (GBMs) are high-grade malignancies with a poor prognosis. The current standard of care for GBM is maximal surgical resection followed by radiotherapy and chemotherapy. Despite all these treatments, the overall survival is still limited, with a median of 15 months. For patients harboring inoperable GBM, due to the anatomical location of the tumor or poor general condition of the patient, the life expectancy is even worse. The challenge of managing GBM is therefore to improve the local control especially for non-surgical patients. Interstitial photodynamic therapy (iPDT) is a minimally invasive treatment relying on the interaction of light, a photosensitizer and oxygen. In the case of brain tumors, iPDT consists of introducing one or several optical fibers in the tumor area, without large craniotomy, to illuminate the photosensitized tumor cells. It induces necrosis and/or apoptosis of the tumor cells, and it can destruct the tumor vasculature and produces an acute inflammatory response that attracts leukocytes. Interstitial PDT has already been applied in the treatment of brain tumors with very promising results. However, no standardized procedure has emerged from previous studies. Herein, we propose a standardized and reproducible workflow for the clinical application of iPDT to GBM. This workflow, which involves intraoperative imaging, a dedicated treatment planning system (TPS) and robotic assistance for the implantation of stereotactic optical fibers, represents a key step in the deployment of iPDT for the treatment of GBM. This end-to-end procedure has been validated on a phantom in real operating room conditions. The thorough description of a fully integrated iPDT workflow is an essential step forward to a clinical trial to evaluate iPDT in the treatment of GBM.

## 1. Introduction

Glioblastomas (GBMs) are high-grade malignancies (grade IV WHO) and represent the most common primitive brain tumor in adults with an annual incidence of 5/100,000 [1]. These tumors are associated with a poor prognosis and impaired quality of life. The 2-year survival rate is 27% [1]. Current standard of care relies on combined maximal surgical resection of the tumor, radiotherapy and chemotherapy [2]. The extent of resection depends on the tumor infiltration into the surrounding tissue especially into functional areas [3]. Over the past 20 years, innovative techniques have helped to improve the extent of resection. Among them, fluoroguided surgery (FGS), involving a specific photo diagnostic agent preferentially located in the tumor cells (e.g., 5-aminolevulinic acid (5-ALA)) excited at a specific wavelength illumination (e.g., 400 nm blue light exposure), enhanced tumor visualization under fluorescent microscopy [4]. Such fluoroguided surgery improved the rate of total tumor resection from 36% to 65% [5], resulting in a progression-free survival (PFS) and overall survival (OS) prolongation [6]. Despite this armamentarium, GBM recurrence is the rule. It recurs in 85% of cases in the 2 cm around the initial surgical cavity [7]. Furthermore, some patients cannot be operated on, due to their poor autonomy, comorbidities and tumor extension to eloquent or vital areas of the brain. For such patients, GBM prognosis is even worse [8]. The challenge is therefore to optimize the local control for non-surgical GBM.

Photodynamic therapy (PDT) relies on the interaction of light, a photosensitizer (PS) and oxygen. This interaction fosters the transformation of ground state oxygen to reactive oxygen species and radicals, leading to cell death through different pathways, including necrosis and apoptosis [9]. The PDT efficacy depends in particular on the PS penetrance inside the target tissue. An optimal PS penetrance leads to a high positive predictive value of reaching the therapeutic effect inside the target tissue. The tolerance of PDT relies on the PS selectivity, meaning a high differential ratio in PS concentration between target and non-target tissues, to avoid cell death of non-target tissue. PDT treatments have been applied with success to a variety of pathologies, especially precancerous or cancerous diseases. Among them, actinic keratosis [10,11], gastrointestinal dysplasia and esophagus carcinoma [12,13] can be cited. As previously mentioned, brain cancers are not controlled by current treatments, especially high-grade gliomas. For these tumors, two modalities have been experimented with to deliver PDT: intracavitary PDT, especially adapted for surgically accessible lesions, and interstitial PDT for non-surgical lesions. As performed in the INDYGO clinical trial, intracavitary PDT can be delivered through an expandable balloon illumination device insufflated inside the surgical cavity [14,15,16]. It can also be delivered with semiconductor lasers positioned inside the surgical cavity [17]. Intracavitary PDT is an interesting add-on therapy to the current standard of care. It is a well-tolerated treatment with promising oncological outcomes [15,17].

Interstitial PDT (iPDT) could be an option for unresectable GBM, or at the time of tumor recurrence. Indeed, repeat surgical resection at the time of recurrence is feasible in a minority of selected patients harboring favorable prognostic factors [18]. iPDT could open the doors to treatment for a larger number of patients. iPDT consists of introducing optical fibers connected with a laser diode emitter reaching a specified wavelength to activate the photosensitizer inside the tumor without craniotomy and can be considered as a minimally invasive treatment. Regardless of the modality, The PDT pioneers faced several technical limitations, including unspecific PS, unknown light dosimetry and difficulty in delivering the light energy in an appropriate manner. In spite of that, significant therapeutic effects were reported [19,20].

The technical advances in recent decades, such as the improvement of stereotactic tools (robotized arm), medical imaging (intraoperative scanning) and the development of more specific photosensitizers (e.g., protoporphyrin IX (PpIX) induced by 5-ALA) have made iPDT an even more promising treatment. Herein, we propose a dedicated workflow for the clinical application of iPDT to GBM in a standardized and reproducible manner. Based on our experience in the neurosurgical and PDT fields [14,21,22,23,24], we report the use of a specially designed treatment planning system (TPS) and the implantation of stereotactic optical fibers with robotic assistance and intraoperative imaging.

## 2. Method

The iPDT end-to-end procedure reported below is the result of a multidisciplinary team, associating researchers in PDT (SM, LG), algorithmic engineers (ASD, GB) and neurosurgeons specialized in stereotaxis and neuro-oncology (NR, HAL). The iPDT TPS was developed in the INSERM U1189 unit (OncoThai).

### 2.1. Planning Procedure

#### 2.1.1. Brain Imaging

To simulate the whole procedure, from image acquisition to fiber implantation, we used a brain phantom (Brain Simulator, Synaptive Medical Inc., Toronto, ON, Canada). This phantom dedicated to oncological neurosurgery simulation includes a realistic cranial structure, a dura mater containing cerebral fluid inside and a brain harboring tumor-like inclusions, visible on CT or MRI scans. This phantom was used to perform a proof of concept.

The phantom underwent a brain MRI, using a 1.5 Tesla MRI, currently used for clinical practice (General Electric, Boston, MA, USA). The following imaging sequences were performed with one-millimeter slice thickness: 3D T1, axial T2 and 3D FLAIR. In a real patient, 3D T1 after gadolinium infusion would have been additionally performed.

#### 2.1.2. Segmentation Process

The obtained MRI images were uploaded in the Brain Tumor Image Analysis software (BraTumIA) to perform automatic segmentation of healthy and tumor brain tissues [25]. BraTumIA is an open access, validated automatic segmentation tool developed by Bern University. Seven forms of brain tissue were included in the segmentation (cerebro-spinal fluid, white matter, grey matter, enhancing tumor, non-enhancing tumor, perilesional edema and necrosis). The segmentation process is used in clinical practice only. The segmentation was not applied on the phantom because it was not relevant and not necessary to demonstrate the feasibility of the complete procedure.

#### 2.1.3. Host Software for the TPS

The Myrian^®^ software (Intrasense, Montpellier, France) was used to create a dedicated treatment planning system (TPS) for interstitial PDT. We uploaded in the Myrian software the abovementioned MRI sequences (T2, FLAIR, T1 after gadolinium infusion) and the result of the BraTumIA automatic segmentation. Then the tumor (called the “target”) lesion was manually contoured, defining a volumetric region of interest (in cm^3^) further referred to as the target tumor volume (Figure 1A).

#### 2.1.4. Optical Fibers Positioning

The objective of iPDT is to fully overlap the target tumor volume with a “therapeutic dose”. Such optimal overlapping depends on the positioning and on the number of implanted optical fibers. We strictly used cylindrical diffusing optical fibers to illuminate a volume and not a point (contrary to direct shot optical fibers which deliver the light only at their extremity). To place an optical fiber with the TPS, the user selects the entry point and the target point of the fiber (Figure 1B). The stereotactic coordinates of these two points (X (medial lateral), Y (anterior posterior) and Z (cranial caudal)) automatically attributed by the Myrian software are registered and exported. The length of the diffusing part of the optical fiber can be adjusted from 2 to 5 cm (lengths of 2, 2.5, 3, 4 and 5 cm are possible). In case of two or more fibers, the diffusing parts of the fibers have to be at least 9 mm apart [26]. Indeed, for a distance lower than 9 mm, a thermal effect has been reported whereby the light power is too high in presence of PpIX, reaching a temperature higher than the physiological tolerance of the brain parenchyma, that is 43 °C. As a consequence, if the diffusing part of the fibers are closer than 9 mm, our software emits an alert and prevents the user from continuing to the next stage. The software also alerts when two fiber trajectories cross. Once fiber trajectories and lengths are defined, a dedicated algorithm using Monte Carlo simulation developed in-house (OncoThai Laboratory) calculates the effective volume treated with iPDT. The process of simulation and calculation is described in detail in the section below.

#### 2.1.5. Laser Devices

Dedicated laser devices were developed in the laboratory. Each of these laser devices integrates two diode drivers and has two addressable fiber output channels with same wavelength of 635 nm. These devices can deliver power up to 2 watts for each output. Each output can be controlled separately to optimize the delivered dose. A touch screen interface was included for a simple treatment flow.

#### 2.1.6. Monte Carlo Simulations

In order to ensure that the number, lengths and positions of the cylindrical diffusing optical fibers positioned by the neurosurgeon inside the brain allow the treatment of the entire manually contoured target volume, 3D Monte Carlo simulations were performed. These simulations are based on the “mcxyz” program developed by Jacques et al. to model light transport in a heterogeneous medium consisting of different types of tissues with varying optical properties (absorption coefficient, scattering coefficient, anisotropy factor and refractive index) [27]. These simulations were performed including the optical parameters of the abovementioned seven brain tissues segmented using BraTumIA. The photosensitizer concentration in the tissues was also included. An effective treated volume was estimated from these simulations and then compared to the manually contoured target through sensitivity analysis. Specificity analysis was not required due to the safety of the treatment for healthy tissues. Indeed, in high-grade tumors, the concentration ratio of PpIX is up to 100:1 between the tumor and the healthy parenchyma [28]. Thus, selectivity is obtained by the respective PpIX concentration and not by the laser illumination.

##### Specifications of the “Mcxyz” Program for Interstitial 5-ALA PDT

The “mcxyz” program requires some specifications before being run.

(a)Specification of the wavelength of interest

As our TPS was designed for interstitial 5-ALA PDT using 635 nm homemade light sources (ONCOTHAI, Lille, France), the simulations were performed using the optical properties at 635 nm. This wavelength, corresponding to the Q_I_ absorption band (i.e., to the fifth most intense absorption band) of PpIX [29], promotes light penetration in biological tissues while ensuring efficient excitation of molecular oxygen to its singlet state.

(b)Specification of the light source power

For the simulation, a light source power of 200 mW/cm was considered whatever the length of the selected cylindrical diffusing fibers, as previously reported in the literature [20,26].

(c)Specification of the 3D Cartesian grid of voxels

The “mcxyz” program requires a 3D grid of voxels. Here, this 3D grid of voxels simply consists of the voxels of the T1-weighted MRI images of the patient. The spatial resolution therefore is set to that of these T1-weighted MRI images.

(d)Specification of the optical properties of the 3D grid of voxels

Each voxel of the 3D grid is associated with the optical properties at 635 nm of the tissue it was assigned to with BraTumIA. To determine the optical properties at 635 nm for the seven tissue types (i.e., enhancing tumor, non-enhancing tumor, necrosis, oedema, white matter, grey matter, cerebrospinal fluid), we performed a broad literature review [30,31,32,33,34] (Table 1). For each parameter, we used the median value reported in the reviewed publications.

(e)Specification of the sampling method for the initial position and propagation direction of the photon packets

The “mcxyz” program requires a large number of photon packets to be launched in the abovementioned 3D Cartesian grid of voxels. The initial position and direction of propagation of each photon packet therefore need to be set.

The three options for setting the initial position of the photon packet in the “mcxyz” program (manually set launch, uniform beam and isotropic point source) are not directly applicable for 5-ALA iPDT. In fact, as mentioned, 5-ALA iPDT mainly involves light from cylindrical diffusing fibers. The different cylindrical diffusing fibers planned to be used during the treatment (2, 2.5, 3, 4 and 5 cm) were implemented in the TPS as isotropic line sources of different lengths. For each length, the probability density function defining the distribution of the initial position of the photon packets was derived from the corresponding normalized light emission profile we measured. Based on the characteristics of the profiles of all the cylindrical diffusing fibers, super-Gaussian probability density functions were used. Parameters of these super-Gaussian probability density functions were obtained by fitting and were used to generate an initial position for the photon packet. In order to test our treatment planning software, we imported anonymized data from real patients harboring glioblastomas. This helped us to improve the segmentation process on complex targets. No real patient was treated with iPDT in this study. Figure 2 shows the light emission profile of the 2 cm cylindrical diffusing fiber and the associated super-Gaussian probability density function.

Distribution of the 100,000 initial generated positions sampled from this probability density function is also reported in Figure 3. Regarding the setting of the propagation direction, a sampling according to an isotropic distribution was applied.

(f)Specification of the number of photon packets and use of the parallelization

Due to the complexity of our simulations, at least than 100,000 photon packets are required to achieve relevant and reproducible results. Sequential consideration of such a number of photon packets in the “mcxyz” program leads to huge computational time incompatible with the constraints of a TPS. To overcome this limitation, we adopted parallel programming on graphics cards. The “mcxyz” program therefore is a graphics processing unit (GPU) implemented on an NVIDIA GPU Quadro K620 using the Compute Unified Device Architecture (CUDA) C programming language. Such parallel programming promotes the use of a power of two for the number of photon packets. Based on a trade-off between computational time and high-quality simulations, the number of photon packets was set to 131,072.

##### Definition of the Effective Treated Volume

Given the abovementioned specifications, the “mcxyz” program was run for each inserted cylindrical diffusing fiber positioned by the neurosurgeon. Once all these runs were completed, each voxel of the 3D grid was associated with as many fluence rates (mW/cm^2^) as there were inserted cylindrical diffusing fibers. Summing these fluence rates provided an overall fluence rate (mW/cm^2^) for each voxel. Multiplying this overall fluence rate by the treatment time allows the fluence (J/cm^2^) in each voxel to be obtained. The effective treated volume by PDT was then defined as the voxels with a fluence higher than 25 J/cm^2^, which was previously defined as the minimal target fluence [14,15]. The volume of this effective treated volume is displayed in cubic centimeters on the TPS (Figure 3). The coverage index corresponding to the percentage of the manually contoured target volume overlapped by the effective treated volume (i.e., treated by PDT) is also displayed. The user can eventually adjust the planification to improve the coverage index, according to the match or mismatch between the manually contoured target volume and the effective treated volume obtained after fiber placement. Once the treatment is approved, the software generates a report with the stereotactic coordinates of the optical fibers.

### 2.2. Surgical Procedure

In the neurosurgical operative room, the phantom was fixed with a rigid head holder (Mayfield, Integra, Cincinnati, OH, USA), which was itself fixed to the robot (Figure 4 and Figure 5). The optical fibers’ stereotactic coordinates generated by the TPS were implemented in the neurosurgical planning software (neuroinspire^TM^, Renishaw, United-Kingdom) of the surgical robot (Neuromate, Renishaw). We adapted the neurolocate^TM^ module which enables intraoperative X-ray/CT registration without the need for bone or skin anchored fiducials (Figure 3). Once the robot was in place, a 3D scan was performed, using an intraoperative cone beam CT (CBCT) scan (Oarm, Medtronic) (Figure 5A). The acquired skull and brain 3D scan was co-registered with preoperative brain MRI. This co-registration allows the robot to move according to the previously defined stereotactic coordinates of the entry point of each respective fiber, respecting the alignment between the entry point and the target point (for each fiber: entry point and target point, defining an axis). Once the robot was in position, the skull was drilled at the target entry point. In a real patient, we would coagulate with a thin monopolar probe the dura mater and the cortical entry point to avoid subsequent bleeding due to the fiber guide insertion. Then, the fiber guide was introduced through the working channel held by the robotic arm to the target point (Figure 5B), using the distance between the target and the robotic arm as the z coordinate reference (z distance, e.g., 130 cm). Then the robotic arm was removed and replaced with the fiber guide. The optical fiber was introduced inside a transparent fiber guide, taking into account the length of the diffusive part of the optical fiber. As the fiber guide is closed at its extremity, it secures the optical fiber insertion, avoiding going too deep. The abovementioned steps were repeated for each optical fiber to settle.

Once every fiber was in position, an intraoperative CBCT scan was performed to check the accuracy of the fiber placement. We uploaded the CBCT scan images to the robot workstation and coregistered them with the preimplantation images with apparent fiber trajectories (these trajectories were drawn from the optical fibers’ stereotactic coordinates implemented in the software of the robot). Performing this coregistration allowed a strict and reproducible positioning check of the optical fibers (Figure 6). After the positioning was checked, the optical fibers were connected to the laser devices developed in the OncoThai unit. Once the treatment was performed, the fiber guides and the optical fibers were removed. In real patients, the exiting skin points of each fiber would have been sutured (with only one stitch needed for each fiber).

## 3. Discussion

The interstitial PDT has many strengths that should improve the management of glioblastomas. iPDT could be an add-on therapy for unresectable GBM and/or at the time of tumor recurrence. First, iPDT is a minimally invasive treatment which is performed without craniotomy and without dissection of brain parenchyma. As a consequence, iPDT reduces the risk of morbidity. Lietke et al. recently reported their clinical experience of over 44 patients who underwent iPDT for GBM recurrences [35]. In their series, only one patient (2%) reported a new neurological deficit still present 6 weeks after treatment. All other reported deficits resulting from brain swelling or small bleeding were transient. Second, regarding its low invasiveness, one could assume that iPDT could be repeated on demand, depending on the GBM evolution. iPDT could focus on areas of the tumor recurrence showing contrast enhancement after gadolinium infusion. Third, optical fibers can reach brain areas that cannot be dissected, such as basal ganglia (e.g., thalamus) or the brainstem. Fourth, the selective therapeutic effects of iPDT help preserve the functional bundles in the eloquent areas, unlike laser interstitial thermal therapy (LITT) which alters all tissue in the target area. Indeed, the LITT does not require any PS and relies only on the thermal effect which is not cell selective. Fifth, iPDT appears as an adapted therapy for frail patients or those who do not fill the criteria for an invasive brain procedure. In fact, even if the iPDT procedure is performed under general anesthesia, it should be shorter (approximately 2 h of intervention) than open surgery and enable a faster recovery. Thus, iPDT affords patients harboring a “non-surgical GBM” a new treatment option whether in the case of de novo or recurrent tumors.

For the development of our TPS, we leveraged our neurosurgical expertise, especially in radiosurgery and in stereotactic procedures such as stereo electroencephalography or deep brain stimulation. The abovementioned iPDT TPS relies on target volume delineation whether on T1 after gadolinium infusion or FLAIR images. Our TPS works automatically in a stereotactic referential, i.e., each point on the screen is defined according to three coordinates (x, y, z). These coordinates can subsequently be exported to other applications.

As a result of the planification, we report the volume treated by iPDT and the coverage index. The use of a specific photosensitizer preferentially located in the tumor cells makes iPDT selective. It follows that illumination of healthy tissues without a photosensitizer does not induce parenchyma alteration. Therefore, there is sense in planning a larger illumination area than the target one in order to reach a total coverage of the target (i.e., a coverage index of 100%).

It is of most interest to standardize and optimize the iPDT procedures for GBM. Available planning software on the market (e.g., Target, Brainlab, Munich, Germany) aims at defining a target volume and the optical fiber placement but does not provide the estimated treated volume with iPDT [35]. Our TPS is innovative in the sense that it calculates the estimated treated volume after the optical fiber placement. Our algorithm based on Monte Carlo simulations incorporates the optical parameters of all the brain tissues and helps positioning the fiber placement inside the lesion to achieve an optimal coverage. It also helps avoid the “hot spot” between fibers by respecting a minimal distance of 7–9 mm between each diffusing part [26]. One of the key points in the development of this algorithm was to determine the optical parameters for each brain tissue (Table 1). It required a systematic review. The consistency of these parameters was validated by mathematical simulation but also in the TPS interface.

One could assume that in vivo validation of the TPS should be performed. Although our laboratory has strong experience and expertise in preclinical research [22,23], some limitations of in vivo testing can be reported in our case: (1) the difficulty of reproducing the complexity of the GBM patterns using cell lines in an animal model (e.g., U87 cells do not induce necrosis), (2) the small size of rodent brains that are currently used for such experiments, (3) the variation of the optical parameters between human and other species and (4) the ethical component, which must avoid all nonessential animal experimentation. Lastly, iPDT has been already performed in human clinical practice [35].

One major aspect of iPDT efficacy and tolerance relies on the level of energy delivered to the adjacent tissues. Regarding this level, available literature is sparse and heterogeneous. The fluence (in J/cm^2^) delivered during human iPDT varied from 32 to 1870 J/cm^2^ [20,36]. A tendency towards higher fluence emerges in the recent iPDT series [20,26,37,38], partly due to the use of more selective photosensitizers, which reduce the occurrence of post-treatment brain swelling. In previous preclinical studies, we reported that a dose of 25 J/cm^2^ induces a therapeutic effect, producing tumor deaths through inflammatory (necrosis) and non-inflammatory (apoptosis) processes [22,23]. However, these findings were obtained with a direct-fire optical fiber (non-diffusing one). In our TPS, we used this value of 25 J/cm^2^ as the therapeutic dose, and we simulated the volume receiving a dose equal or higher to the therapeutic one (Figure 6). Using the abovementioned settings (therapeutic dose, optical properties), an optical fiber with a 3 cm diffusing part covers an effective volume treated with iPDT around 1 cm^3^.

In the literature, the number of inserted fibers was up to 6 [20,26,39]. In order to reduce the number of inserted fibers, higher fluence could be investigated. However, it has to be understood that delivering a higher fluence with the same light source power will significantly lengthen the whole procedure. As the procedure is performed under general anesthesia, we consider that an illumination time of 1 h is adequate, while knowing 1 h more is needed for robotic fiber insertion and intraoperative imaging control.

Intraoperative spectroscopy has been advocated for monitoring the PpIX concentration inside the target tissues when performing PpIX induced by 5-ALA iPDT [35]. As the therapeutic effect depends on the presence of PpIX, it could be of interest to pursue the illumination as long as PpIX is present. However, measuring the local PpIX concentration in a reproducible manner is complex and depends on the distance between fibers and the presence of artifacts masking the true signal. As our TPS also takes into account the PpIX concentration depending on the brain tissues, we did not consider intraoperative spectroscopy in the operative workflow.

To our knowledge, all human iPDT series have used a continuous illumination pattern [35,40]. We reported in previous preclinical studies the interest of light fractionation to let the target tissue reoxygenate between illumination periods [22,23,24]. A 2 min interval is enough to reach pre-iPDT O2 concentration in the treated tissue [41]. We also advocate for 100% O2 ventilation during the whole procedure. Using a fractionated light scheme, it increases the non-inflammatory response, inducing selective tumor cell death through the apoptotic pathway, and it reduces the inflammatory response with less necrosis and peripheral edema. As reported in early post-iPDT MRI, the therapeutic effect of treatment is also enhanced using light fractionation [23]. The perfusion index is elevated in the surrounding tumor area, which corresponds to the opening of the blood–brain barrier, fostering the immune system response and the efficacy of adjuvant treatment such as chemotherapy or immunotherapy. This finding illustrates the potential synergy of iPDT with current oncological treatments.

Another question in iPDT is the target definition. When contouring the target in brain MRI, we usually stick to the peripheral ring of enhancements on the T1 sequence after gadolinium infusion. This corresponds to the proliferating border of the tumor. The glioblastoma cells, at a lower proportion, are located up to 1 to 2 cm away from the tumor bulge. This infiltrative pattern is the reason for inevitable tumor recurrence, mainly after 12 months, despite complete initial gross tumor resection, followed by concomitant radio-chemotherapy in the best cases. It would be therefore questionable to extend the target volume to the peripheral edema (on the T2 Flair sequence) in order to catch these “isolated” tumor cells and reduce the recurrence rate. Previous iPDT studies reported intratumoral fiber positioning [20,26,38]. The question is whether it is more efficient to set the fibers all around the tumor bulge instead of inside the tumor (potentially inside the necrosis, with no effect due to the absence of oxygen). Krishnamurthy and al. reported a series of 18 patients treated with PDT using a non-selective PS (hematoporphyrin derivative agent), placing one fiber in the core of the tumor and up to 6 in the periphery. Their response rate was 94% with promising oncological outcomes. Nevertheless, Krishnamurthy reported a significant rate of neurological deficits (28%). This finding highlights the risks of using a non-selective PS and also the risk of peripheral fiber placement, with the inherent damage to the functional neurological bundle.

From a surgical and technical point of view, the workflow depicted herein is also innovative. We report a stereotactic fiber implantation using robotic guidance, without any stereotactic frame. Indeed, the use of a stereotactic frame, with a rigid grid to guide the fiber insertion, hampers the potential flexibility of iPDT. The teams using the rigid frame were compelled to insert their fibers in an orthogonal way [20,26]. As a consequence, some of the tumor area could not be reached with the diffusing part of the fibers. In our protocol, the use of a robotic arm is helpful in increasing stereotactic fiber insertion and affords a variety of entry point to reach any area of the brain. However, the whole procedure could also be performed using a non-robotic system.

Although our iPDT procedure has not yet been performed in clinical practice, we could express some considerations about postoperative care based on our PDT and neurosurgical experience. Regarding the patient management after iPDT, it should not differ from actual surgical treatment. The administration of corticosteroids should be considered in case of a headache or neurological deficit, in order to reduce the potential post-iPDT edema. Postoperative brain MRI should be performed to assess the effect of iPDT, such as contrast enhancement changing, edema variation and perfusion modification around the treated site. Potential bleeding due to the optical fiber insertion may change optical parameters of the target tissue and consequently affect the efficacy of the iPDT. However, such bleeding cannot be predicted and can be seen/assessed only in postoperative imaging (CT scan or MRI). The hospital stay could be 3–4 days, reducing the hospitalization cost. The patients treated with iPDT could benefit from adjuvant treatments, such as radio-chemotherapy, even earlier than patients with open craniotomy due to the absence of extensive scarring.

## 4. Conclusions

The interstitial photodynamic therapy combines several qualities to improve the management of GBM. It is a selective minimally invasive technique with promising oncological outcomes and low morbidity. iPDT remains a potential option for deep-seated tumors in patients with high surgical risks and for tumor recurrence. In such patients, the advantage of an effective treatment volume encompassing the solid tumor volume to reach infiltrative portion of the tumor is pertinent. The integrated workflow reported herein helps optimize the whole iPDT procedure with an objective of standardization and reproducibility. This includes the use of MRI brain imaging, a dedicated TPS taking into account the optical parameters of the brain tissues and the robotic-assisted implantation of optical fibers with intraoperative 3D imaging control. The thorough description of a fully integrated iPDT workflow is a step forward to a clinical trial to evaluate iPDT in the treatment of glioblastomas. Although the dosimetry aspect could still be improved, a clinical evaluation is the next step in bringing iPDT into the current clinical practice.

## Figures and Tables

**Figure 1 cancers-13-05754-f001:**
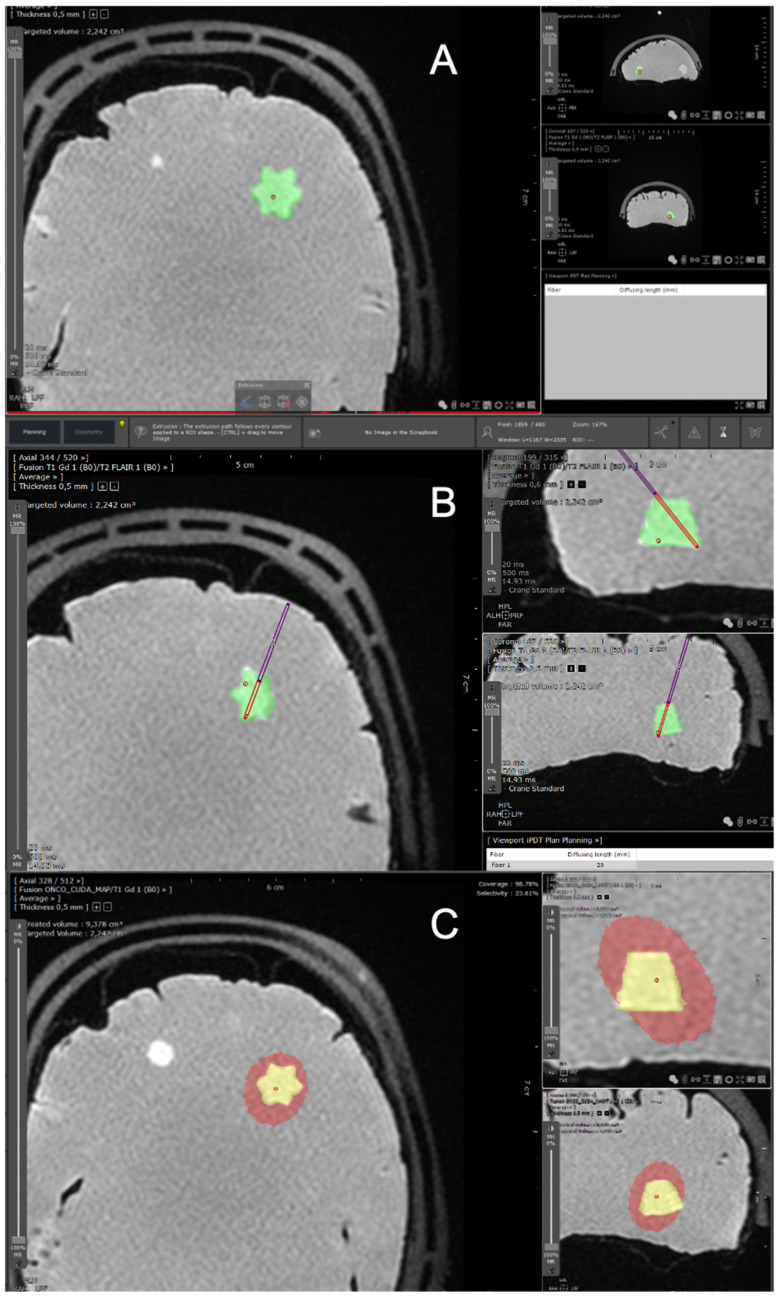
Screenshots from the TPS dedicated to iPDT of GBMs. These screenshots were obtained using a brain phantom (Brain Simulator, Synaptive Resection 750). (**A**) The first step consists of manually contouring the target volume, which is displayed in green (cm^3^). (**B**) Then, an optical fiber trajectory is defined, with an entry point (point of entry into the skull) and a target point (end of the introduced fiber) harboring stereotactic coordinates. The red portion of the fiber corresponds to the diffusing part. (**C**) The resulting effective volume treated with iPDT is displayed in red. The objective is for the red volume to optimally overlap the target tumor volume in yellow. TPS: treatment planning software. iPDT: interstitial photodynamic therapy. GBMs: glioblastomas.

**Figure 2 cancers-13-05754-f002:**
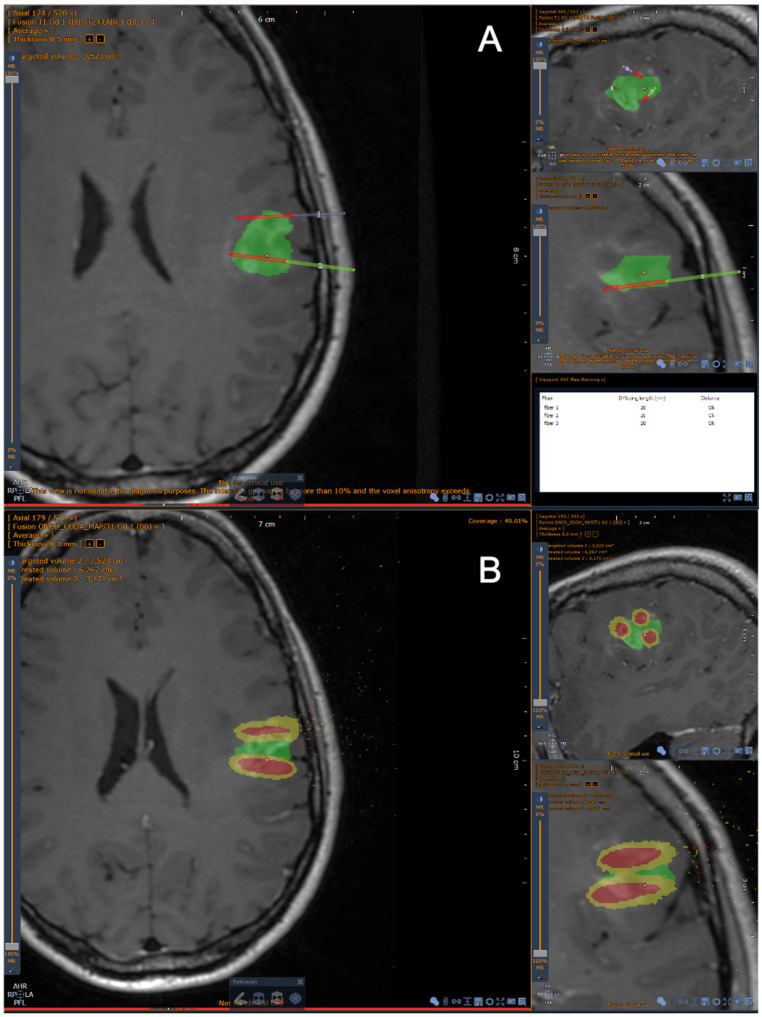
Screenshots from the iPDT TPS for GBM. (**A**) Brain MRI, T1 after gadolinium infusion. The green zone corresponds to the target volume. The diffusing part of each fiber is displayed in red. (**B**) The effective treated volume is displayed in yellow, receiving at least 25 J/cm^2^. The red volume around each fiber corresponds to a volume receiving an energy >250 J/cm^2^. In this snapshot, the optical fibers have a 2 cm diffusing part. Only one fiber is able to cover a volume superior to 1 cm^3^ (here: 1.166 cm^3^, taking into account the optical parameters of all brain tissues).

**Figure 3 cancers-13-05754-f003:**
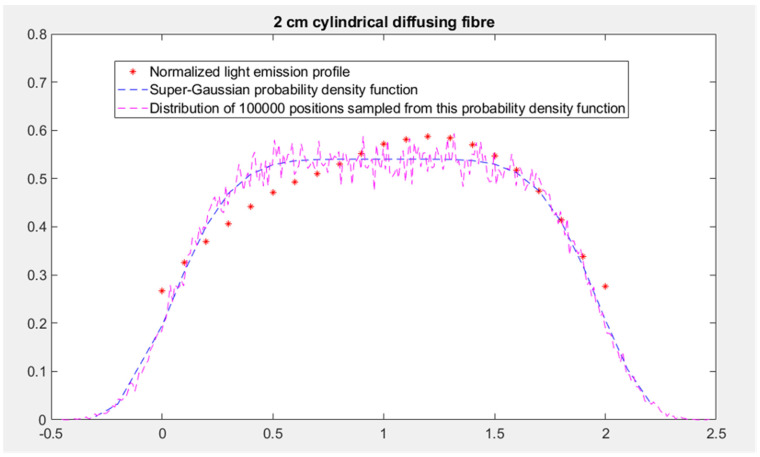
The light emission profile of the 2 cm cylindrical diffusing fiber (represented by red dots), the associated super-Gaussian probability density function (represented by blue dashes) and the distribution of 100,000 positions sampled from this probability density function (represented by magenta dashes).

**Figure 4 cancers-13-05754-f004:**
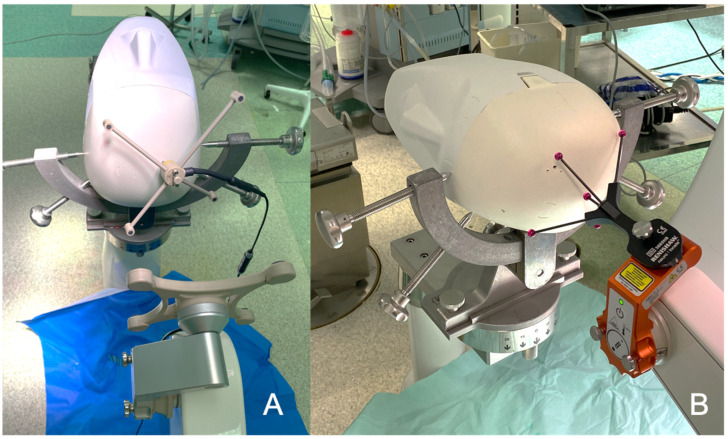
The brain phantom is fixed in the rigid head holder. (**A**) A rigid fiducial is fixed on the bone. Then, an ultrasound registration is performed. (**B**) The registration was performed without any bone anchored fiducial, using the neurolocate^TM^ module adapted on the robotic arm.

**Figure 5 cancers-13-05754-f005:**
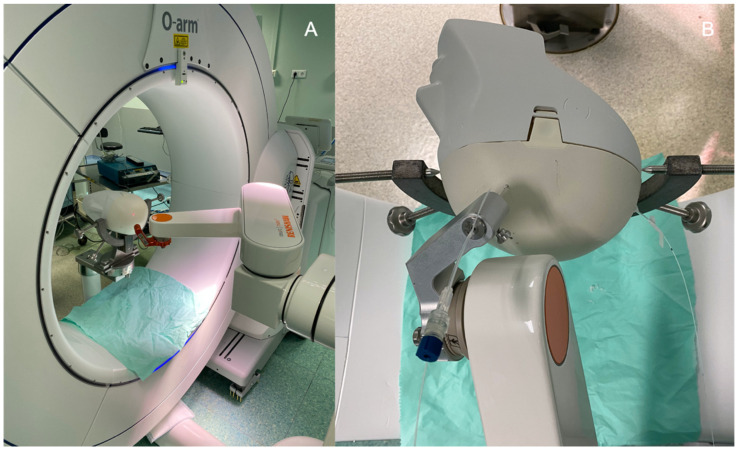
(**A**) The brain phantom is fixed in the rigid head holder. The CBCT (O-Arm, Medtronic) was performed to obtain the 3D images needed to navigate the robotic arm. The laser at the distal part of the robotic arm is pointing at the skull entry point. (**B**) A guide was previously introduced into the brain to ensure the right trajectory before inserting the appropriate optical fiber.

**Figure 6 cancers-13-05754-f006:**
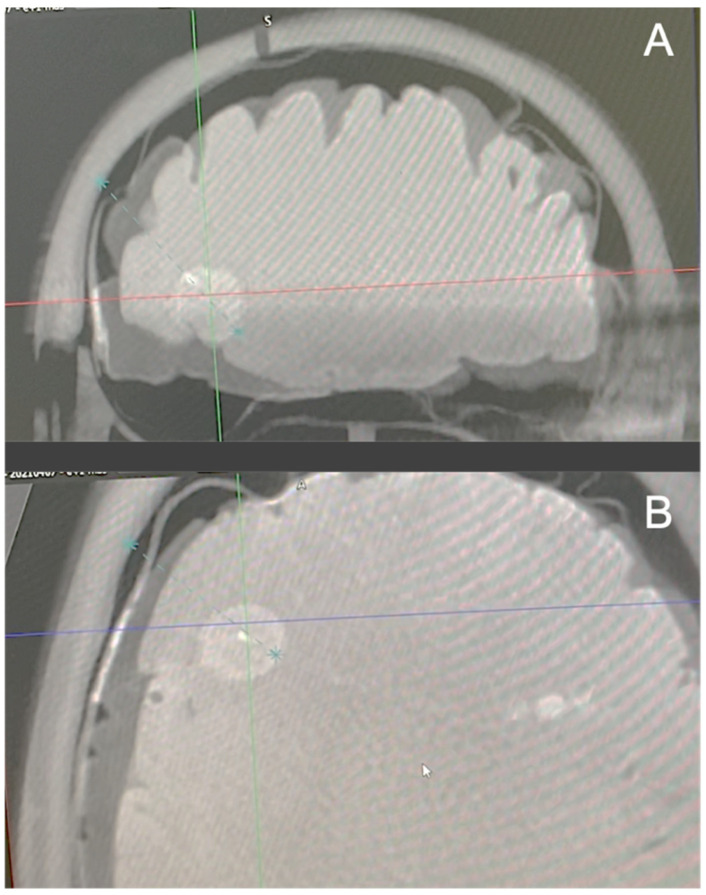
A CBCT (O-Arm, Medtronic) was performed after the insertion of the optical fiber through its guide. The planned trajectory is the dotted blue line. The optical fiber appears in white. As the optical fiber is superposed to the blue line, and the accuracy was estimated as satisfying. (**A**) Coronal plane. (**B**) Axial plane.

**Table 1 cancers-13-05754-t001:** Optical properties at 635 nm for the seven brain tissues considered in the study (issued from references [30,31,32,33,34]). The refractive index (RI) was the same for all subtypes of cerebral tissue (RI = 1.40).

Variable	Healthy Parenchyma			Tumor			
	CSF	Gray Matter	White Matter	Necrotic	Non-Enhancing	Enhancing	Oedema
Absorption coefficient (/mm)	0.004	0.13	0.08	0.17	0.08	0.17	0.08
Scattering coefficient (/mm)	0.009	9	40.5	24.1	69.7	24.1	40.5
Anisotropy factor	0.89	0.92	0.85	0.9	0.9	0.9	0.85

## Data Availability

Data are contained within the article.

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
