# Peer review of "Interstitial Photodynamic Therapy for Glioblastomas: A Standardized Procedure for Clinical Use"

_cancers, 2021, doi:10.3390/cancers13225754_

Round 1

Reviewer 1 Report

The authors describe in their paper a pre-clinical model of interstitial photodynamic therapy (iPDT) using a phantom and a workflow associating intraoperative imaging, treatment planning system and robotic assistance for laser sources intratumoral implantation. The aim of the study is to offer a standardized surgical procedure of iPDT that would be applicable for clinical practice in patients harbouring inoperable cerebral glioblastomas.

The study design and methodology are appropriate and well described, in the streamline of the expertise of the authors in the field and a continuation of their previous reports on PDT. Technological refinements of the iPDT, such as stereotactic placement of laser light sources, a light dosimetry taken into account different compartments of brain tissue and heterogeneous tumors, and light fractionation, would certainly enhance the safety and efficacy of iPDT for brain tumors. The discussion considers appropriately the major issues of the iPDT and refers to pertinent literature.

There are some points in the paper that are unclear and others that deserve discussion.

The major part of the paper concerns the light dosimetry, which certainly is the key point of iPDT. As the authors point out, GBM are highly heterogeneous, complicating the light distribution to the different compartments of the tumor. In the paragraph 2.1.2 (segmentations process), the authors include seven “brain tissues” in the segmentation, which correspond to different compartments of the brain and glial tumors. Did they differentiate those compartments in the phantom of this study, or will this segmentation process be used in patients in clinical practice only ? Figures 1 and 6 do not show these different compartments. This point could be clarified.

Conversely, Figure 3 contains images of a real patient, who is not described in the text. Was he or she included in the present study ? If so, the authors should provide more clinical information, including inform consent and details of the procedure (illumination parameters, tumor compartments description, and if possible postoperative imaging, that would add a pertinent data to the study.

Unwanted side effects such as brain swelling and bleeding may change optical parameters of the target tissue and consequently the efficacy of the iPDT. How the authors would manage these effects ?

Paragraph 2.2 (surgical procedure) “As the fiber guide is closed at its extremity, it secures the optical fiber insertion, avoiding going too deep” : one understands that the whole diffusing tip is inserted within the fiber guide, which should be transparent to allow light to penetrate the target tissue : this point is unclear in the text.

Table 1 gives the absorption and scattering coefficient as well as anisotropy factors of different components of healthy parenchyma and tumor ; how were these factors determined ? were they obtained from previous studies of the authors in real patients, or from literature data ? This point is unclear in the text.

The presented work is based on a phantom study ; therefore, considerations about patients management (as described in the last paragraph of the discussion chapter) are putative and not yet based on a clinical experience reported in this paper.

The conclusion appears optimistic when stating that iPDT could be accessible to a wide range of patients harboring non-surgical glioblastomas. Many of those appear large and widespread on imaging, involving more than one lobe, and even demonstrating a contralateral extension. Those tumors would be particularly difficult to treat with iPDT, requiring multiple light source fibers, increased illumination spots and fluence, with increased risk of side-effects and lower efficacy. On the other side, “small” glioblastomas are most often candidates for surgical resection. Although, iPDT remain a potential option for deep-seated tumors, in patients with high surgical risks, and for tumor recurrence. In those, the advantage of an effective treatment volume encompassing the solid tumor volume to reach infiltrative portion of the tumor, is pertinent but further clinical studies are necessary to confirm the efficacy of iPDT in these patients.

References 33 and 34 are incomplete (authors, editor, publication year, pages, are missing).

Author Response

Reviewer #1:

Thank you very much for your valuable comments and suggestions, which helped us improve this manuscript. Please find below our detailed reply to your comments presented highlighted in yellow.

The authors describe in their paper a pre-clinical model of interstitial photodynamic therapy (iPDT) using a phantom and a workflow associating intraoperative imaging, treatment planning system and robotic assistance for laser sources intratumoral implantation. The aim of the study is to offer a standardized surgical procedure of iPDT that would be applicable for clinical practice in patients harboring inoperable cerebral glioblastomas.

The study design and methodology are appropriate and well described, in the streamline of the expertise of the authors in the field and a continuation of their previous reports on PDT. Technological refinements of the iPDT, such as stereotactic placement of laser light sources, a light dosimetry taken into account different compartments of brain tissue and heterogeneous tumors, and light fractionation, would certainly enhance the safety and efficacy of iPDT for brain tumors. The discussion considers appropriately the major issues of the iPDT and refers to pertinent literature.

We thank reviewer#1 for this comment.

There are some points in the paper that are unclear and others that deserve discussion.

The major part of the paper concerns the light dosimetry, which certainly is the key point of iPDT. As the authors point out, GBM are highly heterogeneous, complicating the light distribution to the different compartments of the tumor. In the paragraph 2.1.2 (segmentations process), the authors include seven “brain tissues” in the segmentation, which correspond to different compartments of the brain and glial tumors. Did they differentiate those compartments in the phantom of this study, or will this segmentation process be used in patients in clinical practice only ? Figures 1 and 6 do not show these different compartments. This point could be clarified.

We agree with this comment of the reviewer#1 and we clarified this point.

The segmentation process will be used in clinical practice only. Indeed, even realistic, the brain phantom does not reproduce such heterogeneity of structures and materials. The objective of using such a phantom was mainly to assess the feasibility of the whole treatment procedure from imaging acquisition to robotic optical fiber implantation under realistic conditions in the operating room.

We added in the paragraph 2.1.2: “The segmentation process will be used in clinical practice only. The segmentation was not applied on the phantom because it was not relevant and not necessary to demonstrate the feasibility of the complete procedure.

Conversely, Figure 3 contains images of a real patient, who is not described in the text. Was he or she included in the present study ? If so, the authors should provide more clinical information, including inform consent and details of the procedure (illumination parameters, tumor compartments description, and if possible postoperative imaging, that would add a pertinent data to the study.

We fully agree with this comment and we added in the 2.1.6.1 section e):

“In order to test our treatment planning software, we imported anonymized data from real patients harboring glioblastomas. It helped us improving the segmentation process on complex targets. No real patient was treated with iPDT in this study.”

Unwanted side effects such as brain swelling and bleeding may change optical parameters of the target tissue and consequently the efficacy of the iPDT. How the authors would manage these effects ?

Regarding brain edema, the presence of pretreatment peritumoral edema is taken into account in the 7-tissue segmentation process (Bratumia). In case of increased brain swelling after iPDT, it could be evaluated on postoperative imaging and justify brief corticosteroid treatment. A potential bleeding due to the optical fibers insertion may change optical parameters of the target tissue and consequently affect the efficacy of the iPDT. However, such bleeding could not be predicted and could be seen/assessed only on postoperative imaging (CT scan or MRI). A phase I clinical study evaluating the treatment tolerance could peculiarly assess the severity of post iPDT edema and the frequence of post fiber insertion bleeding.

We added in the last paragraph of discussion

“A potential bleeding due to the optical fibers insertion may change optical parameters of the target tissue and consequently affect the efficacy of the iPDT. However, such bleeding could not be predicted and could be seen/assessed only on postoperative imaging (CT scan or MRI).”

Paragraph 2.2 (surgical procedure) “As the fiber guide is closed at its extremity, it secures the optical fiber insertion, avoiding going too deep” : one understands that the whole diffusing tip is inserted within the fiber guide, which should be transparent to allow light to penetrate the target tissue : this point is unclear in the text.

We fully agree with the reviewer that this point was not clear. This has been addressed in the 2.2 section:

« The optical fiber is introduced inside a transparent fiber guide, taking into account the length of the diffusive part of the optical fiber. As the fiber guide is closed at its extremity, it secures the optical fiber insertion, avoiding going too deep ».

Table 1 gives the absorption and scattering coefficient as well as anisotropy factors of different components of healthy parenchyma and tumor ; how were these factors determined ? were they obtained from previous studies of the authors in real patients, or from literature data ? This point is unclear in the text.

We obtain these values from previous literature as mentioned in the section 2.1.6.1, d). Unfortunately, Table 1 has been misplaced during editing. We repositioned it.

“To determine the optical properties at 635 nm for the seven tissue types (i.e., enhancing tumor, non-enhancing tumor, necrosis, oedema, white matter, grey matter, cerebrospinal fluid), we performed a broad literature review [30–34]. For each parameter, we used the median value reported in the reviewed publications (Table 1).”

The presented work is based on a phantom study ; therefore, considerations about patients management (as described in the last paragraph of the discussion chapter) are putative and not yet based on a clinical experience reported in this paper.

We fully agree with this comment and we added the following precision in the corresponding paragraph:

“Although our iPDT procedure has not yet been performed in clinical practice, we could emit some considerations about postoperative care based on our PDT and neurosurgical experience.”

The conclusion appears optimistic when stating that iPDT could be accessible to a wide range of patients harboring non-surgical glioblastomas. Many of those appear large and widespread on imaging, involving more than one lobe, and even demonstrating a contralateral extension. Those tumors would be particularly difficult to treat with iPDT, requiring multiple light source fibers, increased illumination spots and fluence, with increased risk of side-effects and lower efficacy. On the other side, “small” glioblastomas are most often candidates for surgical resection. Although, iPDT remain a potential option for deep-seated tumors, in patients with high surgical risks, and for tumor recurrence. In those, the advantage of an effective treatment volume encompassing the solid tumor volume to reach infiltrative portion of the tumor, is pertinent but further clinical studies are necessary to confirm the efficacy of iPDT in these patients.

We fully agree with Reviewer#1 comment. Our conclusion has been rephrased accordingly.

“iPDT remains a potential option for deep-seated tumors in patients with high surgical risks and for tumor recurrence. In those, the advantage of an effective treatment volume encompassing the solid tumor volume to reach infiltrative portion of the tumor is pertinent.”

References 33 and 34 are incomplete (authors, editor, publication year, pages, are missing).

We agree with the reviewer. The reference have been updated:

“33. Yaroslavsky, A.N.; Schulze, P.C.; Yaroslavsky, I.V.; Schober, R.; Ulrich, F.; Schwarzmaier, H.-J. Optical Properties of Selected Native and Coagulated Human Brain Tissues in Vitro in the Visible and near Infrared Spectral Range. Phys Med Biol 2002, 47, 2059–2073, doi:10.1088/0031-9155/47/12/305.

“34. Tuchin, V.V. Tissue Optics and Photonics: Biological Tissue Structures. J Biomed Photonics Eng 2015, 1, 3–21, doi:10.18287/jbpe-2015-1-1-3. »

Reviewer 2 Report

I thank the authors for this interesting manuscript on a promising novel minimally invasive therapy for some glioblastoma patients. The description of a standardised workflow for iPDT constitutes a valuable contribution to the literature. Specifically, their TPS constitutes an important step forward as compared to currently available target delineation software packages.

I have some minor questions / remarks:

  • Do the authors think there exists a dose-effect relationship for iPDT of 5-ALA? Should the standard dose for fluorescent guided surgery suffice also for iPDT?
  • The authors describe the use of a robotized arm for implantation of the optical fibers. Since this device is not yet standard neurosurgical armamentarium, would it also be feasible to use a non-robotic system, e.g. analogous to the Medtronic Vertek® system for frameless stereotactic biopsy?
  • On page 13, the authors state that their workflow obliterates the use of "any frame". I respectfully disagree, since they have to use a head holder. Of course, this boils down to semantics: "frame" vs "head holder". Therefore, I would suggest adding "stereotactic" to the aformentioned sentence: "... robotic guidance, without any stereotactic frame".
  • p 5. §2.1.6. "il" should be "is"

Author Response

Reviewer #2:

We would like to thank Reviewer#2 for the valuable comments and constructive suggestions, which led to this improved version of the manuscript. Please find below our detailed replies to your comments presented highlighted in yellow.

I thank the authors for this interesting manuscript on a promising novel minimally invasive therapy for some glioblastoma patients. The description of a standardised workflow for iPDT constitutes a valuable contribution to the literature. Specifically, their TPS constitutes an important step forward as compared to currently available target delineation software packages.

We thank reviewer#2 for this comment.

I have some minor questions / remarks:

Do the authors think there exists a dose-effect relationship for iPDT of 5-ALA?

We thank reviewer#2 for this relevant question.

There is no stricto sensu dose effect on the efficacy of 5-ALA iPDT as previously reported in our literature review (Leroy, H.-A.; Guérin, L.; Lecomte, F.; Baert, G.; Vignion, A.-S.; Mordon, S.; Reyns, N. Is Interstitial Photodynamic Therapy for Brain Tumors Ready for Clinical Practice? A Systematic Review. Photodiagn Photodyn 2021, 36, 102492, doi:10.1016/j.pdpdt.2021.102492.).

The therapeutical effect of 5-ALA iPDT relies on the presence of oxygen, red light and PpIX.

First, the presence or the absence of PpIX (especially its selectivity and concentration) determines/conditions the iPDT effect. Second, the laser light emission has to be sufficient to go through the tissue and reach deep located cancer cells. However, too strong illumination fosters post iPDT brain swelling when using a non-selective photosensitizer.

Should the standard dose for fluorescent guided surgery (FGS) suffice also for iPDT?

We thank reviewer#2 for this relevant question.

When performing FGS with 5-ALA (Gliolan), we use a blue light (410 nm wavelength) which in fact corresponds to one of the absorption peaks of PpIX-5ALA. At 410 nm, the penetrance of the blue light into the brain tissues is low (inferior to red light) and the energy delivered through the microscope lamp is not sufficient to activate the metabolic cascade of PDT. As a consequence, FGS has no PDT effect.

The authors describe the use of a robotized arm for implantation of the optical fibers. Since this device is not yet standard neurosurgical armamentarium, would it also be feasible to use a non-robotic system, e.g. analogous to the Medtronic Vertek® system for frameless stereotactic biopsy?

We thank reviewer#2 for this relevant question.

The described procedure could actually be performed with a non-robotic system. However, the whole workflow requires a stereotactic environment (a frame, or as mentioned another frameless tool).

The following sentence has been added in the discussion:

However, the whole procedure could also be performed using a non-robotic system. »

On page 13, the authors state that their workflow obliterates the use of "any frame". I respectfully disagree, since they have to use a head holder. Of course, this boils down to semantics: "frame" vs "head holder". Therefore, I would suggest adding "stereotactic" to the aformentioned sentence: "... robotic guidance, without any stereotactic frame".

We agree with this comment and corrected the sentence accordingly.

"... robotic guidance, without any stereotactic frame".

p 5. §2.1.6. "il" should be "is"

We fully agree with the reviewer. The corresponding sentence has been corrected.

“The photosensitizer concentration in the tissues is also included.”
